# Using Micro- and Macro-Level Network Metrics Unveils Top Communicative Gene Modules in Psoriasis

**DOI:** 10.3390/genes11080914

**Published:** 2020-08-10

**Authors:** Reyhaneh Naderi, Homa Saadati Mollaei, Arne Elofsson, Saman Hosseini Ashtiani

**Affiliations:** 1Department of Artificial Intelligence and Robotics, Faculty of Computer Engineering, Iran University of Science and Technology, Tehran 1684613114, Iran; r_naderi@alumni.iust.ac.ir; 2Department of Advanced Sciences and Technology, Islamic Azad University Tehran Medical Sciences, Tehran 1916893813, Iran; h.saadati1995@gmail.com; 3Department of Biochemistry and Biophysics and Science for Life Laboratory, Stockholm University, 106 91 Stockholm, Sweden; arne.elofsson@dbb.su.se

**Keywords:** psoriasis, network analysis, microarray gene expression analysis, combination therapy, modularity

## Abstract

(1) Background: Psoriasis is a multifactorial chronic inflammatory disorder of the skin, with significant morbidity, characterized by hyperproliferation of the epidermis. Even though psoriasis’ etiology is not fully understood, it is believed to be multifactorial, with numerous key components. (2) Methods: In order to cast light on the complex molecular interactions in psoriasis vulgaris at both protein–protein interactions and transcriptomics levels, we studied a set of microarray gene expression analyses consisting of 170 paired lesional and non-lesional samples. Afterwards, a network analysis was conducted on the protein–protein interaction network of differentially expressed genes based on micro- and macro-level network metrics at a systemic level standpoint. (3) Results: We found 17 top communicative genes, all of which were experimentally proven to be pivotal in psoriasis, which were identified in two modules, namely the cell cycle and immune system. Intra- and inter-gene interaction subnetworks from the top communicative genes might provide further insight into the corresponding characteristic interactions. (4) Conclusions: Potential gene combinations for therapeutic/diagnostics purposes were identified. Moreover, our proposed workflow could be of interest to a broader range of future biological network analysis studies.

## 1. Introduction

Psoriasis is a multifactorial chronic inflammatory disorder of the skin, affecting around 2–3% of the population worldwide, which is characterized by hyperproliferation of the epidermis. The most prevalent subtype of the disease is psoriasis vulgaris (plaque psoriasis). Even though psoriasis’ etiology is not fully understood, it is believed to be multifactorial, with numerous key components including genetic susceptibility and environmental triggers in combination with skin barrier disruption and immune dysfunction. Regarding the pathogenesis and the molecular interactions responsible for psoriasis, many details are still unclear. There are a host of treatments for moderate to severe psoriasis patients, ranging from topical creams to systematic drugs and phototherapy. However, our inability to thoroughly understand the disease prevents the optimal use of these therapeutic routes and there is no satisfactory treatment for most psoriatic patients yet [1,2]. Several computational studies have tried to find and elucidate potential contributing genes in psoriasis. Genes that show considerable network interconnections are normally involved in similar or identical biological pathways. Therefore, modularity analysis has become an important part of gene expression network studies. Network analysis can be used to mine interconnections between large amounts of data from biological and medical science. Such methods can give rise to new hypotheses and potential knowledge that would not be easily decrypted otherwise. The characterization of biological networks by means of network topological properties is widely used for gaining insight into the global and local interaction properties [3]. Li et al. performed weighted gene co-expression network analysis (WGCNA) on RNA-Seq data of psoriatic and normal skin tissues to detect the key long non-coding RNAs (lncRNAs) and mRNAs associated with psoriasis. They recognized important lncRNAs in the lncRNA–mRNA co-expression network based on the degree distribution of the network [4]. Zhang et al. analyzed microarray data from plaque psoriasis samples, revealing that metabolic and viral infection-associated pathways were enriched the most. The hub genes, which were identified to be implicated in playing key regulatory roles in the protein–protein interaction (PPI) network, were determined based on the overlapping outcomes achieved by maximal clique centrality (MCC) and density of maximum neighborhood component (DMNC) topological analysis methods [5]. Mei et al. used microarray data in psoriatic patients to construct co-expression modules based on the differential expression of various disease-characteristic genes and screened the likely effective therapeutic molecules and their possible target. Furthermore, the degree centrality measure was used to identify the network hub genes [6].

Sundarrajan et al. conducted an integrated systems biology approach to understand the molecular alliance of psoriasis with its corresponding comorbidities. A microarray dataset was fetched for each disease for this study. “Network medicine” was applied in order to scrutinize the molecular intricacy of psoriasis, leading to the identification of new molecular associations among apparently distinct clinical manifestations [7]. Arga et al. used topological, modular and a novel correlation analysis based on fold changes of the microarray data fetched from three different platforms. Psoriasis-associated PPI networks were studied based on a dual-metric approach using degree and betweenness [8]. Piruzian et al. applied a number of network and meta-analysis techniques to paired lesional and healthy microarray datasets to reveal the similarities and dissimilarities between proteomics- and transcriptomics-level perturbations in psoriasis. They particularly tried to reveal novel regulatory pathways in psoriasis development and progression [9].

In this study, we performed a network topological investigation of highly communicative genes underlying medium-severe psoriasis. We looked into this disease at a systemic level from a network-based perspective in order to investigate the patterns of gene interactions in the context of network and subnetwork analysis results. Our main goal was to determine potentially influential genes that play important roles in psoriasis. The identification of such genes along with their interactions, which is essential for the development of future diagnostic/therapeutic candidates through medical experiments, was also addressed. To the best of our knowledge, this is the first study to consider as many micro- and macro-level network metrics as possible in order to explore the most important long- and short-range interactions of the PPI network in moderate to severe psoriasis vulgaris. Besides the above, we reported the enriched pathways associated with the identified genes to provide further insight into the pathogenesis of psoriasis.

## 2. Materials and Methods

A complete workflow of our study design is shown in Figure 1.

### 2.1. Dataset

The microarray dataset used in this paper was generated by Suárez-Farinas et al. [10] based on the GPL570 Affymetrix Human Genome U133 Plus 2.0 array platform and is available in NCBI’s Gene Expression Omnibus (GEO) repository, with accession number GSE30999. It was selected from 85 patients with moderate to severe psoriasis. The patients had not received any active psoriasis therapy. The samples are defined as NL (non-lesional) and LL (lesional) skin. Considering the sufficiently large number of 170 psoriasis vulgaris paired samples (greater than 30 according to the central limit theorem) in comparison with other available paired tissue datasets, the mentioned dataset was chosen. We eschewed combining different array experiments data from the NCBI’s GEO series in order to avoid the variations associated with batch effects from merging different datasets of multiple microarray experiments and involving different population groups, particularly because psoriasis is a multifactorial disease with a combination of key environmental and genetic factors [11].

### 2.2. Preprocessing

We performed some Quality Control (QC) (Appendix A) checks involving sample-level plots. These checks comprise a Principal Component Analysis (PCA) plot, density function plot, heatmap of Pearson’s correlation coefficient (r2) between samples and boxplot. All pairwise Pearson correlations on samples were calculated by the QC tool accessible from Expression Console software, version 1.4 (Affymetrix). PCA helps us to distinguish samples using expression variations and determine whether the LL and NL samples are differentiable after normalization (Figure 2). The gene expression levels were transformed to logarithmic base 2 scale. We applied MAS 5.0 algorithm [11] implemented by Affymetrix, available in the affy package in Bioconductor. The preprocessing was performed using R (version 3.6.1).

### 2.3. Selection of Differentially Expressed Genes (DEGs) in Psoriasis Vulgaris Patients

The DEGs between NL and LL samples were assessed using a moderated *t*-test called the empirical Bayesian method [12]. The proportion parameter of genes for the eBayes algorithm was considered 0.01. The cutoff criteria of *p*-value < 0.01, adjusted *p*-value (FDR) < 0.01 and |logFC| ≥ 1 were considered as the thresholds for the significance to extract DEGs among 54,675 probe sets. The top 2000 genes (Appendix A) were selected in order to be submitted to the STRING database [13]. The hgu133plus2.db annotation package [14] was used to transform probe IDs into gene symbols. For one gene symbol corresponding to several probe IDs, the maximum absolute value of logFC for probes was measured as the final value. All the procedures up to this point was processed under the R statistical environment (version 3.6.1) with the use of the package limma in Bioconductor release 3.9.

### 2.4. PPI Undirected Unweighted Network Reconstruction of DEGs

To reconstruct the network of PPI data, we used STRING database version 10.5, covering more than 2000 organisms, including physical as well as functional associations. DEGs were submitted to STRING with minimum required interaction score 0.4 and confident score ≥0.15 [13].

### 2.5. Primary Communicative Genes Identification

We topologically analyzed the psoriasis-associated PPI network imported from STRING as undirected and unweighted. A total 1481 nodes and 11,704 edges were detected using NetworkAnalyzer [15], which is a Java plugin integrated into network software tool Cytoscape 3.7.2 [16]. The influence of the genes in the network was characterized and quantified based on their position in the network, which was described using centrality measures, including local scale (degree) and global scales (stress, betweenness, closeness and eigenvector). We considered them as micro-level network metrics. In network science, the mentioned metrics are directly proportional to the topological importance of the nodes [17]. More than 10 percent of the total number of 1481 nodes, being derived with respect to the mentioned metrics, were called primary communicative genes. As a result, minimum thresholds of degree 50, stress 200,000 and eigenvector 0.05 were considered. By the union of three gene sets, 152 genes called primary communicative genes were obtained. Eigenvector metric was calculated by CentiScaPe plugin [18].

### 2.6. PPI Network Reconstruction and Analysis of Primary Communicative Genes

The primary communicative genes were identified according to the interactions between the nodes associated with the 1445-node connected network. We hypothesized that these genes could be a subset of influential nodes within the mentioned network; hence, the dependencies of these genes to themselves could be more likely to be important than their dependencies to other nodes in the 1445-node network. Therefore, we merely considered the relationships between these 152 genes themselves after mapping them to STRING. The resulting network was analyzed with the metrics, namely degree, stress, betweenness, eigenvector and closeness, using Cytoscape 3.7.2 [16] (Appendix A).

### 2.7. Macro-Level Network Analysis Followed by Top Communicative Genes Identification

Modularity is one of the topological features of interconnected nodes evaluating the quality of the modules, resulting from modularity detection techniques in the context of network science. The modularity detection problem considers decomposing a network into modules of densely connected nodes [19]. To further topologically characterize the PPI network of primary communicative genes, we used a modularity detection algorithm based on a heuristic method proposed in [19] that uses modularity optimization. We tried to maximize modularity to gain the highest density modules possible (Figure 4b). The micro-level network metrics obtained from the analysis of the primary communicative genes’ PPI network, mentioned in the previous section, were then applied to each module to identify top communicative genes. Further investigations into gene sets in each module in terms of metrics showed that some metrics (closeness in modules 1 and 2, eigenvector in module 1) had very low variance in their corresponding values for the nodes, in such a way that those metrics could not be used as discriminating criteria. Consequently, the top communicative genes from each module were extracted with the parameters of degree, stress, betweenness and eigenvector. To further investigate relationships between top communicative genes themselves in each module and between two modules, we extracted the nested subnetworks involved in top communicative genes in each module (intra-module interactions) and between modules (inter-module interactions) (Figure 4c,d). We called the modularity detection step with intra- and inter-module interaction analysis “macro-level network analysis”. Pathway enrichment analysis was performed by web-based tool Enrichr [20] via WikiPathways [21] and Kyoto Encyclopedia of Genes and Genomes (KEGG) 2019 Human databases [22]. We used Gephi (version 0.9.2), an open-source software [23], to apply the modularity detection procedure and Cytoscape to identify and visualize intra- and inter-module interactions.

## 3. Results

### 3.1. Preprocessing of Samples

The PCA plot represents two groups in the dataset after normalization (Figure 2). The Pearson’s correlation coefficient (r2) values as a heatmap indicate a strongly positive correlation between all samples (Appendix A). Appendix A illustrates less sample median fluctuations after normalization.

### 3.2. DEGs in Psoriasis Vulgaris Patients

To visualize DEGs in terms of significance and the magnitude of changes in their gene expression levels as a united plot, we generated a volcano plot (Appendix A). The top 50 upregulated and top 50 downregulated genes in psoriasis-associated DEGs are listed in Appendix A, respectively.

### 3.3. PPI Network of DEGs Reconstruction and Analysis

The PPI network in the STRING database demonstrated 1836 nodes (1481 connected nodes) and 11,704 interactions, including average node degree of 4.8. The network associated with 1481 identified genes was analyzed and reconstructed using Cytoscape (Figure 3). This network contains 18 connected components characterized by one 1445-gene component (network 1), 15 two-gene components (networks 4 to 18) and two three-gene components (networks 2 and 3). The network statistics is demonstrated in Figure 3.

By analyzing micro-level network metrics on component 1 in Figure 3, a total of 152 primary communicative genes were identified among 1481 genes according to combining degree, stress and eigenvector metrics (Figure 4a). Regarding the mentioned micro-level metrics, using the plots of the centrality metrics as illustrated in Appendix A, we examined the pairwise interplay of the metrics in such a way that we gained the greatest number of primary communicative genes, which is the set of genes for the following modularity detection and the subsequent top communicative genes identification.

The ranges of micro-level network metrics and the number of genes for each metric are demonstrated in Table 1.

### 3.4. PPI Network of Primary Communicative Gene Reconstruction and Analysis

The primary communicative genes PPI network consisted of 149 connected nodes, with average node degree of 47, three single nodes (DNAH8, NALCN, PNPLA7) and 3623 edges. The statistics of the 149-gene PPI network are described in Appendix A.

### 3.5. Macro-Level Network Analysis 

Two high density modules were recognized as a result of modularity detection in the 149-connected node PPI network (Figure 4b). Studying the roles of the genes in each module showed that the genes comprising module 1 were closely related to the cell cycle, of which 81 genes were upregulated and five genes were downregulated. On the other hand, module 2 consisted of 63 genes appearing to be most likely relevant to immune system, of which 43 genes were upregulated and 20 genes were downregulated.

### 3.6. Top Communicative Genes Selection 

Appendix A depicts the ranges of all metrics in each module. 

#### 3.6.1. Module 1

Appendix A presents the summary statistics for significant genes selected by filtering micro-level metrics in module 1. Top communicative genes in this module were recognized with a minimum degree of 87, a minimum threshold of 8732 and 0.03 for stress and betweenness metrics, respectively. Ranked values of obtained genes are set out in Table 2. The genes shared among all three groups, including CCNA2, CCNB1 (the highest-ranked value of all gene groups), MKI67, CDK1, CDC20 and FOXM1 as top communicative genes, are shown in subnetwork 1 (Figure 4c).

The main biological functions of top communicative genes in module 1 are demonstrated in Appendix A.

Enriched pathway analysis (module 1)

Among enriched pathways associated with module 1, significant pathways through KEGG and WikiPathways 2019 HUMAN databases are listed in Appendix A.

#### 3.6.2. Module 2

All cutoff criteria of degree ≥ 26, eigenvector ≥ 0.007, stress ≥ 8518 and betweenness ≥ 0.04 were considered as the thresholds for significance to extract top communicative genes in module 2 among 63 genes. The subgraph of relationships between 11 identified top communicative genes is presented in Figure 4c. Appendix A compares summary statistics for module 2 analysis, in which the overlapping genes between all groups are EGF and STAT3. Ranked values of these genes for each metric are presented separately in Table 3.

Main biological functions are revealed for top 11 communicative genes in module 2 in Appendix A.

Enriched pathway analysis (module 2)

Corresponding pathway enrichment results were obtained from WikiPathways and KEGG 2019 HUMAN databases (Appendix A). As can be compared, overlapping pathways between KEGG and WikiPathways databases consist of RIG-I-like receptor signaling pathway, toll-like receptor signaling pathway, AGE-RAGE signaling pathway in diabetic complications and TGF-β signaling pathway.

### 3.7. Identification of Pathways Associated with Genes in Both Modules 1 and 2

Pathway enrichment of combined genes from modules 1 and 2 was performed by KEGG and WikiPathways databases. Subsequently, the top 20 pathways were obtained with the highest combined score. Those pathways containing intersectional genes among modules 1 and 2 were chosen (Appendix A).

### 3.8. Inter Top Communicative Genes Subnetwork Interactions

The subnetwork including the top communicative genes from the two modules is summarized in Figure 4d. These may act as molecular signatures for underlying phenotypes of psoriasis disease. The overall numbers of six and 11 genes (17 in total) for modules 1 and 2, respectively, were identified among a total number of 149 genes as top communicative genes.

## 4. Discussion

Through the reconstruction of a PPI network using DEGs, followed by network analysis, 152 primary communicative genes were chosen from the initial 1481 connected nodes in PPI. As the final step of network analysis, on the basis of the macro-level metrics, the primary communicative genes PPI network was shown to belong to two distinct modules comprising cell cycle (module 1) or immune system-related genes (module 2). Each module was examined with different network micro-level metrics in order to determine the top communicative genes. Module 1 top communicative genes contained CCNA2, CCNB1, CDC20, CDK1, FOXM1 and MKI67; module 2 involved STAT3, EGF, H2AFX, IL1B, IFNG, STAT1, CXCL8, CXCL10, MAPK14, MCL1 and UBE2N (Table 2 and Table 3).

Below, the top communicative genes for which there is experimental evidence relevant to psoriasis are discussed, while mentioning those psoriasis-related intra- and inter-module interactions identified in our study (Figure 4c,d). Considering that the 17 mentioned genes have all been experimentally proven to be important in psoriasis, it is postulated that the other members of these modules could also be related to psoriasis and are worthy of experimental investigation. The cell cycle and immune system genes highlighted in our study as module 1 and 2 are described below.

### 4.1. Module 1 (Cell Cycle)

Psoriatic lesional skin is characterized by a thick epidermis and hyperproliferation of keratinocytes, which is a consequence of keratinocyte reactions to various cytokines made by innate and adaptive immune system responses [24]. CCNB1 had the highest betweenness, degree and stress measures. It was demonstrated by J.L. Melero et al. to have a higher expression in psoriatic tissue together with three other genes, i.e., CCNA2, CCNE2 and CDK1, which could lead to uncontrolled cell proliferation [7]. CDK1 expression is increased in human psoriatic lesions [25]. It can bind to CCNB1 [26] and CCNA2 [27], which have been categorized as module 1 top communicative genes, and CCNB2 [28], which exists in module 1. CCNA2, along with two other cyclins i.e., CCNB1 and CCNB2, involved in cell proliferation, in addition to cell division cycle genes *CDC2* and *CDC20*, are upregulated in psoriatic skin [29]. FOXM1 is one of the core transcription factor regulators in psoriasis [30]. It is the only downregulated top communicative gene. The forkhead box transcription factor is critical for various phases of the cell cycle. FOXM1 has the regulatory role for a group of genes that control the G2/M transition. The malfunction of FoxM1 leads to chromosomal instability and mitotic failure [31]. Michell S. Levin et al. suggested that, in aneuploid cells, immune cells can be recognized through two messengers, i.e., cGAS, a cytosolic DNA sensor, and cGAMP. cGAMP activates the stimulator of interferon genes, resulting in the production of type I interferon and other proinflammatory cytokines that eventually trigger the immune response [32]. CDC20 is an activator of anaphase-promoting complex or cyclosome (APC/C) and plays a critical role in mitosis and S phase through the degradation of S phase and mitosis cyclins, which leads to the exit from mitosis. APC/C phosphorylation by CDK1-cyclin B1 leads to the induced binding of CDC20 to APC/C. The activated $\mathrm{APC}/C^{\mathrm{cdc}20}$ target cyclin B1 APC/C also promotes the start of chromosome segregation in the metaphase–anaphase transition by degrading a protein that inhibits anaphase [33]. APC/C is vital for genomic integrity through the regulation of mitosis. Abnormal APC/C activity leads to genomic instability [34]. The effects of genome instability and the failure of chromosome segregation during the mitosis of the psoriatic cells is not clearly understood and requires more investigation.

### 4.2. Module 2 (Immune System)

EGF exerts its effects through binding to an EGF/TGF α receptor in responsive cells. Any alteration in EGF binding pattern has been shown to lead to abnormal differentiation and growth found in diseases such as psoriasis [35]. Apoptosis inhibition and keratinocyte hyperproliferation are evident in psoriasis. Several proteins are present in these mechanisms and one of them is EGF [36]. EGF is upregulated in lesional psoriatic skin in comparison to non-lesional skin in our results. Since keratinocytes bear receptors for the majority of psoriasis-signature cytokines, they represent the “key responding” tissue cells to the psoriatic microenvironment. They respond to psoriatic cytokines by proliferating and amplifying inflammation through the production of other cytokines (i.e., IL-1F9, TNFα, IL-17C, IL-19, TSLP), chemokines proliferation-stimulating factors and other pro-inflammatory products such as AMPs [37]. STAT3, p-STAT3, UBE2N and CDK6 are downregulated by the overexpression of hsa-miR-4516, which is consistently upregulated in psoriasis and induces apoptosis in HaCaT cells [38]. According to our analysis, STAT3 is upregulated, which explains more severe conditions in lesional relative to non-lesional conditions. IL1B is a key factor in cutaneous inflammation and plays an axial role in the instigation of an inflammatory Th17 micro-milieu in psoriasis and other autoinflammatory diseases. Its biosynthesis is controlled at the transcription level, followed by ensuing posttranslational process by means of inflammatory caspases. Zwicker et al. detected that inflammatory caspase-5 is active in psoriatic skin lesions and epidermal keratinocytes. Besides the above, they showed that IFNG and IL17A cooperatively triggered caspase-5 expression in keratinocyte culture and it was dependent on psoriasis, which is an antimicrobial peptide (S100A7) [34]. IFNG, i.e., single type II IFN, plays a key role in the trigger and escalation of systemic autoimmunity. In conjunction, the interferons affect a wide range of biological responses such as anti-tumor effects, yielding protection against bacterial and viral infections and regulation of effector cells in adaptive and innate immune responses [39]. Using microarray analysis of in-vitro derived macrophages which were treated with IFNG, Fuentes-Duculan et al. showed that a plethora of the genes upregulated in macrophages were present in psoriasis, namely HLA-DR, STAT1, CXCL9 and Mx1 [40]. The importance of IFNG in autoimmune responses is in compliance with its stunningly high overexpression observed in lesional compared to non-lesional samples. CXCL8 is secreted by endothelial cells, monocytes, fibroblasts, macrophages and lymphocytes. This chemokine acts as a bridge between the cell cycle and immunity, leading to the activation of keratinocytes of the lesional skin to produce inflammatory cytokines. It can also participate in the angiogenesis and migration of neutrophils and T cells in the inflammation site. CXCL8 is regulated by other cytokines such as IL17A and IL1B, which is critical for the pathogenesis of psoriasis and influences lesional keratinocytes of psoriasis [41,42,43]. IL12B codes the p40 subunit of IL-23 and IL-12, cytokines playing key roles in Th17 and Th1 procedures, respectively. The alterations in this gene significantly enhance the risk of psoriasis. Johnston et al. demonstrated that a psoriatic risk haplotype at the IL12B locus triggers an increase in the expression of IL12B in monocytes. This increase is correlated with elevated serum levels of IFNG, IL-12 and CXCL10, which is an IFNG-induced chemokine [44]. UBE2N has roles in cellular processes by contributing to the immune system, DNA replication and repair. UBE2N accounts for the ubiquitination of TRAF family, which regulates toll-like receptors on dendritic cells, macrophages and neutrophils in the NFkB pathway of the innate immune system. Therefore, UBE2N could contribute to the inflammatory response [45,46]. H2AFX expression was shown to be elevated in psoriatic arthritis patients compared to control ones through the proteomic analysis of synovial fluid. Therefore, H2AFX is considered as a potential biomarker in ensuing serum investigations [47]. STAT1 expression and activity are both considerably increased in lesional psoriatic skin conditions compared to non-lesional psoriatic skin [48]. MCL1 encodes an anti-apoptotic protein, myeloid cell leukemia-1. Ultraviolet B radiation therapy inhibits growth and induces apoptosis of human skin cells by downregulating MCL1 expression [49]. This gene shows upregulation in our study. MAPK14 and LYN are two instances of protein kinase and signal integrators through which immune cascades are channelized. They are expressed in higher amounts in lesional psoriasis [50].

### 4.3. Pathway Enrichment

In order to achieve a deeper insight into the interactions of primary communicative genes, we performed pathway enrichment analysis. Based on Wikipathways and KEGG pathways enrichment analysis of module 1, the top five enriched pathways from either database were the cell cycle, progesterone-mediated oocyte maturation, oocyte meiosis, p53 signaling pathway, cellular senescence, regulation of sister chromatid separation at the metaphase–anaphase transition, retinoblastoma gene in cancer, gastric cancer network 1 and ATM signaling pathway. Likewise, for module 2 type II interferon signaling (IFNG), IL-10 anti-inflammatory signaling pathway, influenza A, pertussis, hepatitis C, toll-like receptor signaling pathway and RIG-I-like receptor signaling were enriched (Appendix A). Thereafter, based on pathway enrichment analysis of 149 merged genes from module 1 and 2, intersectional pathways with the largest numbers of top communicative genes would be influenza A, cellular senescence, Epstein–Barr virus infection, hepatitis B, photodynamic therapy-induced AP-1 survival signaling and regulation of toll-like receptor signaling pathway (Appendix A).

## 5. Conclusions

In conclusion, despite the existence of numerous medications devised for soothing the symptoms of psoriasis, no ultimate treatment has been achieved for the disease. Evidently, the determination of pivotal genes that play important roles in the pathogenesis of psoriasis is critical for the development of diagnostic/therapeutic targets. Owing to the fact that, out of 17 mentioned genes, all of them have been experimentally proven to be related to psoriasis, we believe that there is a high likelihood that the other members of these modules are also related to psoriasis. Additionally, only one of the top communicative genes, i.e., CXCL8, was among the top 100 differentially expressed genes (Appendix A), indicating that the potentially crucial genes found through our network analysis workflow would not be straightforwardly identified by merely relying on DEGs or without incorporating macro- and micro-level network metrics. Thereby, primary communicative genes found in our study and not investigated elsewhere in connection to psoriasis are put forward for further demonstrations (Appendix A).

Subsequent large-scale validation of the latter in serum is critical in order to prove these proteins as putative biomarkers or/and therapeutic targets for the diagnosis and/or the treatment of psoriasis. Beyond this, since there are several experimental studies in which combinatorial gene sets have been used as therapeutic/diagnostic targets [51,52], from a network science point of view, leading to our intra- and inter-module interactions asserted by experimental studies, it is likely that the mentioned top communicative subnetworks will unveil disease-specific patterns.

## Figures and Tables

**Figure 1 genes-11-00914-f001:**
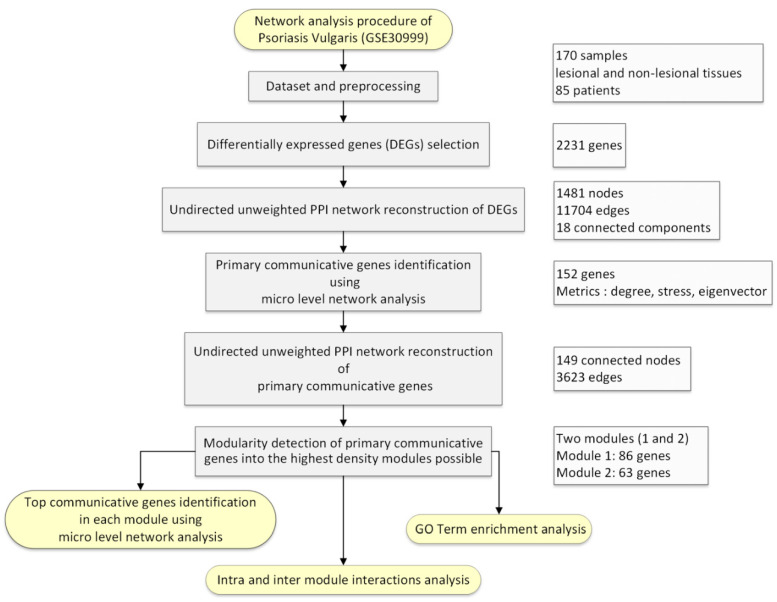
Workflow of the study design of the network analysis of psoriasis vulgaris patients dataset (GSE30999).

**Figure 2 genes-11-00914-f002:**
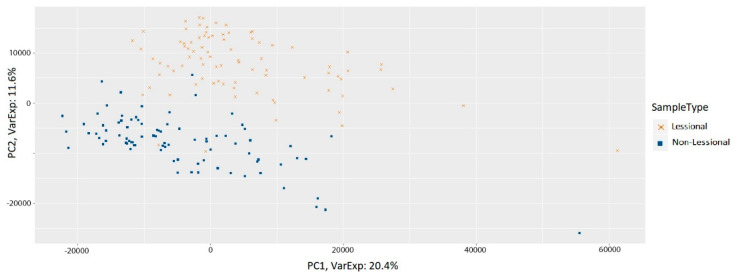
PCA of gene expression values of all samples after normalization. PCA analysis conveys an overall certain discrimination associated with gene expression levels between the two sample types.

**Figure 3 genes-11-00914-f003:**
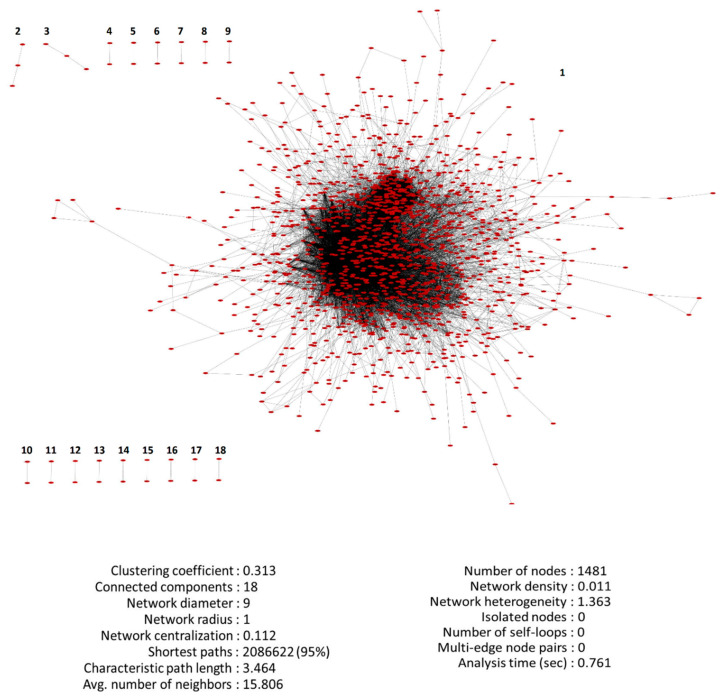
The statistics of 1481-gene PPI network after analysis by Cytoscape.

**Figure 4 genes-11-00914-f004:**
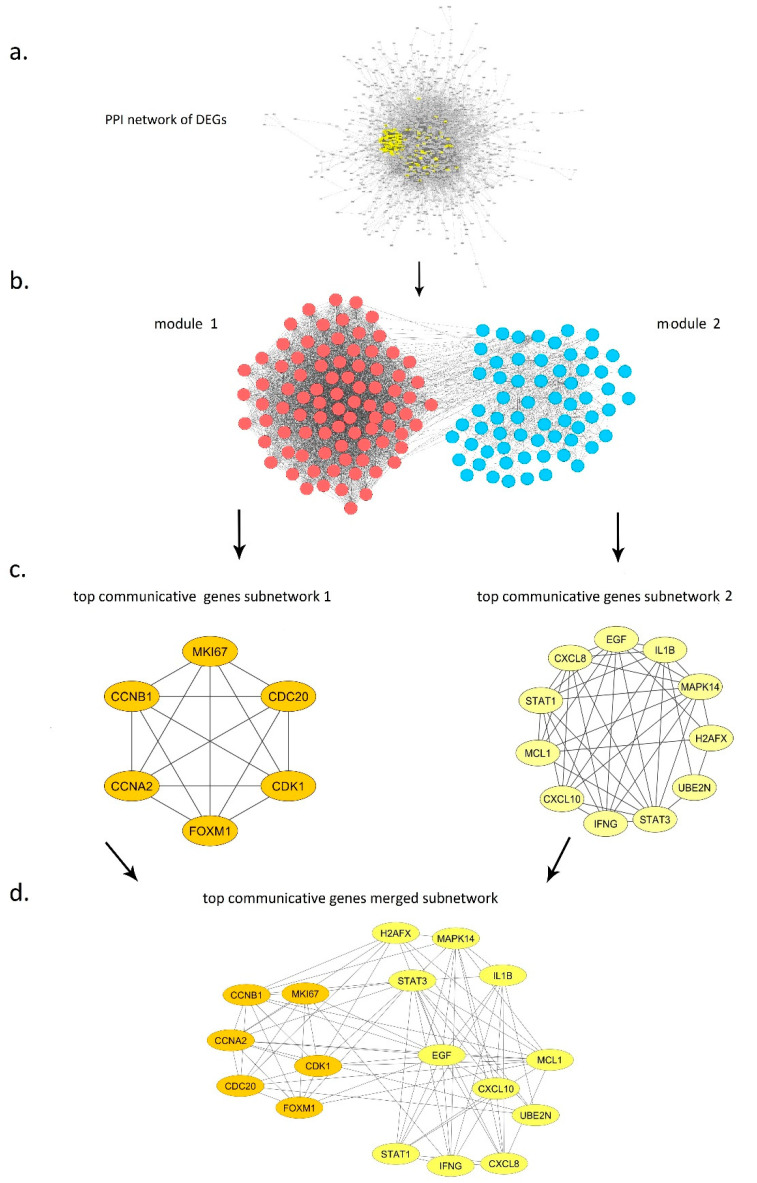
Network analysis results. (**a**) Main component of PPI network of DEGs among 18 components. Selected yellow nodes indicate primary communicative genes based on micro-level network analysis. (**b**) Modularity detection on primary communicative genes PPI network: colors red and blue indicate module 1 and 2 (related to cell cycle and immune system genes), respectively. (**c**) Top communicative genes subnetworks for each module. (**d**) Inter-module top communicative genes subnetwork.

**Table 1 genes-11-00914-t001:** The characteristics of analysis of 1481-gene (all genes) PPI network based on micro-level network metrics.

Micro-Level Network Metrics	Min Value (All Genes)	Max Value (All Genes)	Min Value (Extracted Genes)	Max Value (Extracted Genes)	No. of Extracted Genes
Degree	1	182	50	182	113
Stress	0	2,118,086	200,000	2,118,086	69
Eigenvector	0	0.13	0.05	0.13	86

**Table 2 genes-11-00914-t002:** Top communicative genes (module 1) ranked based on a. degree, b. betweenness, c. stress.

a.			b.		c.	
Rank	Gene Symbol	Degree	Gene Symbol	Betweenness	Gene Symbol	Stress
1	CCNB1	92	CCNB1	0.05	CCNB1	15,636
2	CDC20	91	CDC20	0.04	MKI67	13,412
3	CDK1	90	MKI67	0.04	CDK1	10,020
4	CCNA2	87	FOXM1	0.03	CCNA2	9618
5					FOXM1	8920
6					CDC20	8732

**Table 3 genes-11-00914-t003:** Top communicative genes (module 2) ranked based on a. degree, b. eigenvector, c. stress, d. betweenness.

	a.	b.	c.	d.
Rank	Gene Symbol	Degree	Gene Symbol	Eigenvector	Gene Symbol	Stress	Gene Symbol	Betweenness
1	STAT3	43	H2AFX	0.05	STAT3	29,876	EGF	0.07913
2	H2AFX	43	MCL1	0.018	EGF	27,610	STAT3	0.06636
3	EGF	40	STAT3	0.015	MAPK14	10,910	MAPK14	0.044914
4	IL1B	36	EGF	0.014	IL1B	8518		
5	STAT1	34	UBE2N	0.012				
6	IFNG	31	MAPK14	0.007				
7	CXCL8	31

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
