# Peer review of "Using Micro- and Macro-Level Network Metrics Unveils Top Communicative Gene Modules in Psoriasis"

_genes, 2020, doi:10.3390/genes11080914_

Round 1
Reviewer 1 Report
The Authors have analyzed a large transcriptomic dataset from patients with psoriasis, consisting of paired samples of lesion and non-lesion skin samples, complemented with protein-protein interaction (PPI) data from the STRING database, to identify genes which are important in psoriasis.
The Authors first performed classical identification of differentially expressed genes (DEG) at the mRNA level, followed by selection of nodes with highest network centrality, which they named "micro-level analysis" and the resulting genes - "primary communicative genes". The next step, termed "macro level network analysis", consisted of detection of network modules, followed by re-calculation of centrality measures separately for each module and extraction of sub-networks with highest values of these measures, with the resulting genes named "top-communicative genes". The resulting two modules contained over-representation of cell-cycle genes in module 1 and of immune system genes in module 2. Moreover, all 17 of the top-communicative genes have previously published links to psoriasis.
While the methodology proposed by authors seems promising, with all steps reasonably well described, the study has major problem in its design.
If the aim of the study is to propose a new method "micro- and macro-level network analysis" then this method should be applied alongside other methods to the same or several datasets and their performance compared using several metrics. The comparison to the top 50 DEG is not appropriate, as it does not include the PPI data as input. It is also not clear, how the centrality thresholds used during micro- and macro-level analysis were chosen. Without automation (e.g. using top ranks instead of thresholds), the methodology can hardly be called a "pipeline".
If, on the other hand, the aim of the study was to identify new promising therapeutic targets and interactions in psoriasis, than the importance of the top-communicative genes should be proven beyond the fact that their are already known in psoriasis, preferably by experiments and/or analysis of clinical data.
Author Response
Dear Reviewer1,
All the authors would like to express their wholehearted appreciation for your insightful and sophisticated comments.
We’ve done our best to address every single comment provided by you.
Below, please find the responses and the corresponding changes in the manuscript.
Many thanks for your constructive attention,
Best wishes,
The authors.

Reviewer 2 Report
The manuscript “Using micro- and macro-level network metrics unveils top communicative gene modules in psoriasis” by Naderi et al, sheds a much-needed light on identifying novel targets in pursuit of a cure for psoriasis. The presented results stand to serve as an important template for future investigations and translational science and the authors have done an appreciable work for it. I have nothing to take away or add to this excellent work. I had one query – were there any sex-specific differences in the two modules?
Author Response
Dear Reviewer2,
All the authors would like to express their wholehearted appreciation for your kind, insightful and sophisticated comments.
Below, please find the response to your point.
Many thanks for your constructive attention,
Best wishes,
The authors.

Reviewer 3 Report
Manuscript Reyhaneh Naderi et al. entitled "Using micro- and macro-level network metrics unveils top communicative gene modules in psoriasis" is an interesting research in the field of biomarker searching for psoriasis. Introduction and methodology have been written correctly. The results are presented clearly. However, the discussion is poorly written - it resembles a gene enumeration, with a basic description of their functions and possibly already described changes in gene expression levels. The mechanisms underlying the etiopathogenesis of psoriasis and the association of these genes with the disease have not been described in detail. Interactions between the proposed candidate genes have not been demonstrated and described sufficiently, nor have changes in the levels of expression of these genes been associated with the stage of the disease or therapy (although there are references to it). Discussion of the results obtained in the light of current scientific research (e.g. Coda 2012, Chen 2016, Wang 2019, Zhao 2019 ...) would increase the value of the manuscript. Without validation of the results, it is difficult to conclude about the potential importance of the results obtained for a better understanding of psoriasis. Please also comment on the added value compared to the source research from 8 years ago.
Author Response
Dear Reviewer3,
All the authors would like to express their wholehearted appreciation for your insightful and sophisticated comments.
We’ve done our best to address every single comment provided by you.
Below, please find the responses and the corresponding changes in the manuscript.
Many thanks for your constructive attention,
Best wishes,
The authors.

Round 2
Reviewer 1 Report
Dear Authors,
thank you for the response, and the editing changes which improved the manuscript. I suggest the additional following minor changes:
1. Please include Figure 1 from the response to Reviewer 1 into the manuscript as a supplementary figure.
2. Please describe how the three metrics were combined to be used for selection (e.g. union, intersection, other way)
lines 206-208: "a total of 152 primary communicative genes were identified among 1481 genes according to combining degree, stress and eigenvector metrics" do not provide this detail.
3. Numbering and content of Suppl. Figure with the caption: "Metrics distribution for each module" needs to be corrected.
lines 231-233: "Figure S5 depicts the ranges of all metrics in each module."
The figure with this content, bearing the caption: "Figure S4. Metrics distribution for each module. [...]" is contained in supplementary file: "Figure S4.pdf".
Please also correct the figure, so that the ranges of metrics are better visible, especially for the x-value=0. Also, the x-values should either match or be mapped in the figure legend to the modules' numbers.
4. A label repeated twice, once with item with no corresponding figure file, in the Supplementary Materials list (a leftover from an earlier ms version?)
lines 410-412 "Figure S5: Primary communicative 412 genes PPI network analyzed by Cytoscape. Figure S5: Metrics distribution for each module."
The label "Figure S5" is repeated twice, a figure file with the caption: "Primary communicative 412 genes PPI network analyzed by Cytoscape", is missing.
Author Response
Dear Reviewer1,
Your stunningly constructive and to the point comments were highly effective for a considerable improvement in our manuscript. We are all appreciative of it.
Best wishes,
The authors

Reviewer 3 Report
I thank the authors of the manuscript for their detailed comments. The authors provided comprehensive and accurate replies to the comments submitted. They also made the necessary corrections in the text, which significantly affected the quality of the article. I have no further substantive comments.
Author Response
Dear Reviewer3,
We are highly appreciative of your attention to the details, which has enabled us to considerably improve our manuscript.
Best wishes,
The authors